# Visualizing formation of the active site in the mitochondrial ribosome

**Viswanathan Chandrasekaran[1†], Nirupa Desai[1†], Nicholas O Burton[2,3†], Hanting Yang[1], Jon Price[3,4], Eric A Miska[3,4,5]\*, V Ramakrishnan[1]\***

[1]MRC Laboratory of Molecular Biology, Cambridge, United Kingdom; [2]Centre for Trophoblast Research, Department of Physiology, Development and Neuroscience, University of Cambridge, Cambridge, United Kingdom; [3]Gurdon Institute, University of Cambridge, Cambridge, United Kingdom; [4]Department of Genetics, University of Cambridge, Cambridge, United Kingdom; [5]Wellcome Sanger Institute, Wellcome Genome Campus, Cambridge, United Kingdom

**\*For correspondence:**
eam29@cam.ac.uk (EAM);
ramak@mrc-lmb.cam.ac.uk (VR)

[†]These authors contributed equally to this work

**Competing interests:** The authors declare that no competing interests exist.

**Abstract** Ribosome assembly is an essential and conserved process that is regulated at each step by specific factors. Using cryo-electron microscopy (cryo-EM), we visualize the formation of the conserved peptidyl transferase center (PTC) of the human mitochondrial ribosome. The conserved GTPase GTPBP7 regulates the correct folding of 16S ribosomal RNA (rRNA) helices and ensures 2'-O-methylation of the PTC base U3039. GTPBP7 binds the RNA methyltransferase NSUN4 and MTERF4, which sequester H68-71 of the 16S rRNA and allow biogenesis factors to access the maturing PTC. Mutations that disrupt binding of their *Caenorhabditis elegans* orthologs to the large subunit potently activate mitochondrial stress and cause viability, development, and sterility defects. Next-generation RNA sequencing reveals widespread gene expression changes in these mutant animals that are indicative of mitochondrial stress response activation. We also answer the long-standing question of why NSUN4, but not its enzymatic activity, is indispensable for mitochondrial protein synthesis.

## Introduction

Mitochondrial ribosomes (mitoribosomes) are essential for the production and maintenance of the electron transport chain machinery. As with all other ribosomes, the biogenesis of mitoribosomes involves a complex series of coordinated steps during which the ribosomal RNA (rRNA) folds, proteins are incorporated and the active site is formed. The process is facilitated by specific biogenesis factors, including GTPases, rRNA modifying enzymes, RNA helicases, ribonucleases, chaperones, and other scaffolding and adaptor proteins (*De Silva et al., 2015*). Studying this process in molecular detail is clinically important because defective mitoribosome assembly can lead to encephalomyopathy, optic neuropathy, cardiomyopathy, and hereditary spastic paraplegia, among other disorders.

The general principles of ribosome assembly are well-conserved across species, but mitoribosome assembly has diverged in notable ways from, and involves different factors than bacterial 70S ribosomes. Analogous to bacterial and eukaryotic cytosolic ribosomes, mitoribosome assembly also proceeds via a series of quality-control checkpoints and GTP hydrolysis serves as the commitment step to drive the reaction forward (*Britton, 2009*; *Karbstein, 2007*). Four GTPases, GTPBP5, 6, 7, and 10 (and perhaps GTPBP8; *Maiti et al., 2021*) directly bind the 16S rRNA of the 39S mitoribosomal large subunit (mtLSU) to facilitate its maturation (*Kotani et al., 2013*), but their mechanistic roles and order of action remain poorly understood. Here, we present the cryo-electron microscopy (cryo-EM) structure of the human mtLSU in complex with GTPBP7, an RNA methyltransferase NSUN4 and an RNA scaffolding protein MTERF4, establish the importance of this complex during mitoribosome

biogenesis in vivo and decipher the mechanism by which NSUN4·MTERF4·GTPBP7 ensure maturation of the peptidyl transferase center (PTC) of the mitochondrial ribosome.

## Results

To trap GTPBP7 and other GTPases that participate in human mitochondrial translation, we included the non-hydrolyzable GTP analog β,γ-methyleneguanosine 5′-triphosphate (GMPPCP) during mitoribosome purification. Single-particle cryo-EM (*Figure 1—figure supplement 1*, *Figure 1—figure supplement 2*, *Table 1*) of these and other mitoribosome-bound complexes (*Desai et al., 2020*) also yielded a 5% subset of mtLSU intermediates bound to GTPBP7·GMPPCP in a pre-hydrolysis state and arrested at an assembly state equivalent to the 45S (RI$_{50}$) to 50S transition of the bacterial LSU (*Achila et al., 2012*; *Fahnestock et al., 1973*; *Seffouh et al., 2019*). In this state, the PTC is not completely disordered as it is in PDB 5OOM, an earlier late assembly state (*Brown et al., 2017*), but key helices in the vicinity are held in an unfolded state.

Surprisingly, GTPBP7 was bound to a heterodimer of the 5-methyl cytosine (m$^5$C) modifying enzyme, NOP2/Sun RNA methyltransferase 4 (NSUN4) and its binding partner, mitochondrial transcription termination factor 4 (MTERF4) (*Cámara et al., 2011*; *Figure 1*). NSUN4, MTERF4, and GTPBP7 bind the inter-subunit interface of the mtLSU in a radial arrangement centered about the peptidyl site (P site) and <30 Å from the PTC (*Spåhr et al., 2012*). Additionally, the anti-association factors MALSU1·L0R8F8·mt-ACP are also present on these particles to prevent premature SSU joining (*Brown et al., 2017*; *Desai et al., 2020*).

The NSUN4 active site contains weak density for the methyl donor S-adenosyl methionine (SAM). Moreover, the cryo-EM density suggests that there is no m$^5$C methylation of 16S rRNA located within ~20 Å of the NSUN4 active site indicating that methyltransferase activity is not involved at the mtLSU assembly checkpoint. Bisulfite sequencing studies across many species have also not revealed any m$^5$C targets on the 16S rRNA, making this an unlikely site of NSUN4 methyltransferase activity (*Metodiev et al., 2014*).

Our structure represents an intermediate in which the proteins uL16m, bL27m, and bL36m have already bound, but before complete folding of rRNA domain IV helices 68–71 and dissociation of NSUN4, MTERF4, and GTPBP7 by GTP hydrolysis, and subsequent small subunit (mtSSU) joining. The 16S rRNA domain IV helices 68–71 (nucleotides 2542–2637) typically span the A, P, and E sites and pack against the D-loop, and anticodon arms of the P-site tRNA (*Figure 2*). Here, they are instead held in a partially unfolded state by MTERF4 at a location that would permit biogenesis factors to access to their binding sites, but clash with an incoming SSU (*Figure 2C*). Subsequent GTP hydrolysis by GTPBP7 would result in dissociation of the GTPase and MTERF4·NSUN4, allowing H68–71 to fold (compare *Figure 2C,D*).

RbgA, the bacterial homolog of GTPBP7, facilitates the incorporation of the mitoribosomal protein bL36m into the mtLSU (*Britton, 2009*; *Kotani et al., 2013*; *Figure 2—figure supplement 1*). On the other hand, two other ribosomal proteins uL16m and bL27m, whose incorporation in bacteria is also attributed to RbgA, are already present in an earlier assembly intermediate (PDB 5OOM). We suspect that this difference is likely because only two GTPases (ObgE being the other) regulate bacterial 50S assembly while at least four are implicated in ribosome assembly in human mitochondria (*Maiti et al., 2021*). Remarkably, GTPBP7 binds in the vicinity of 2′-O-methylated bases in the A and P loops of the PTC, which are critical rRNA elements that bind the aminoacyl and peptidyl tRNAs, respectively. Our structure reveals that the A and P loops move toward GTPBP7 and NSUN4, respectively, relative to their positions in the mature 55S (*Figure 2—figure supplement 2*) and His 34 of helix 1 of GTPBP7 directly contacts the highly conserved U3039 to verify 2′-O-methylation by mitochondrial methyltransferase 2 (MRM2) (*Figure 2B*; *Lee and Bogenhagen, 2014*). Given the universal importance of 2′-O-methylation of specific nucleotides in the PTC for translation (*Pintard et al., 2002*; *Rorbach et al., 2014*), it is striking that GTPBP7 directly contacts Um3039. In principle, slight rearrangements of helix 1 of GTPBP7 could also ensure G3040 methylation by MRM3. Similarly, G2815 (in the P loop), which is 2′-O-methylated by MRM1 (*Lee and Bogenhagen, 2014*; *Rorbach et al., 2014*) is in the vicinity of NSUN4 and GTPBP7 in our structure (*Figure 2—figure supplement 2*). We propose that GTPBP7 ensures 2′-O-methylation of these residues by coupling GTP hydrolysis and subsequent GTPBP7 egress from the 39S to its interaction with the methylated bases.

**Table 1.** Data collection, processing, refinement, and model statistics.

| | |
|---|---|
| **Data collection** | |
| Voltage (kV) | 300 |
| Pixel size (Å) | 1.04 |
| Detector | Falcon III |
| Defocus range (μm) | −1.1 to −3.2 |
| Defocus mean (μm) | −2.1 |
| Electron dose (e$^-$ frame$^{-1}$ Å$^{-2}$) | 1.5 |
| **Data processing** | |
| Independent data collections | 7 |
| Useable micrographs | 43,950 |
| Particles picked | 3,374,367 |
| Final particles | 66,340 |
| Map sharpening B-factor (Å$^2$) | −10 |
| Resolution (Å) | 3.4 |
| EMPIAR accession code | EMPIAR-10809 |
| EMDB accession code | EMD-13329 |
| PDB accession code | 7PD3 |
| **Model composition** | |
| Chains | 70 |
| Non-hydrogen atoms | 108,053 |
| Protein residues | 9276 |
| RNA bases | 1522 |
| Metals (Mg$^{2+}$/Zn$^{2+}$) | 91/2 |
| Ligands | GCP (1), SAM (1) |
| **Refinement** | |
| Resolution (Å) | 3.4 |
| CC (mask) | 0.70 |
| **R.M.S deviations** | |
| Bond lengths (Å) | 0.009 |
| Bond angles (°) | 1.068 |
| **Validation** | |
| Molprobity score | 1.66 |
| Clashscore, all atoms | 4.97 |
| Rotamers outliers (%) | 0.01 |
| Cβ outliers (%) | 0.01 |
| **Ramachandran plot** | |
| Favored (%) | 94.17 |
| Allowed (%) | 5.69 |
| Outliers (%) | 0.14 |

GTPBP7, MTERF4, and NSUN4 are highly conserved and their counterparts in *Caenorhabditis elegans* are MTG-1 (33% identical), MTER-4 (21%), and NSUN-4 (34%), respectively. In *C. elegans*, the methyltransferase activity of NSUN4 for mitochondrial protein synthesis was shown to be dispensable (*Navarro et al., 2021*), suggesting that its importance may instead lie in its interaction with the conserved core of the mtLSU. To test the importance of complex formation of the three protein factors with the mtLSU during assembly in vivo, we used *C. elegans* as a model organism and

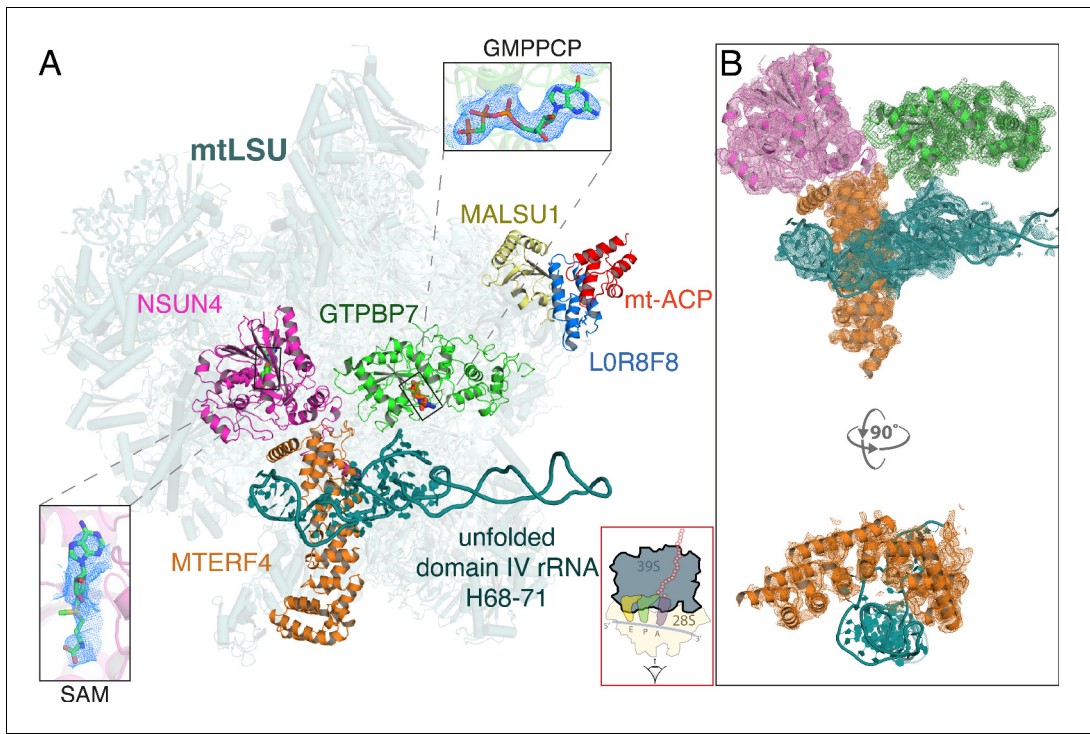

**Figure 1.** Architecture of the mitoribosomal LSU complexed with NSUN4·MTERF4·GTPBP7·GMPPCP and MALSU1-L0R8F8-mt-ACP. (**A**) The inter-subunit interface faces the reader and a cartoon of an elongating mitoribosome is shown in the red box (bottom) to aid orientation. NSUN4 (pink) bound to S-adenosyl methionine (SAM, left inset), GTPBP7 (green) bound to the non-hydrolyzable GTP analog, GMPPCP (top inset) and MTERF4 (orange) interact with the subunit interface of the mtLSU assembly intermediate. The domain IV rRNA helices H68-71 (teal) are unfolded, rendering the aminoacyl, peptidyl, and exit sites incomplete. Also pictured are the anti-association factors MALSU1 (yellow), L0R8F8 (blue), and mt-ACP (red). (**B**) Map-to-model fits for NSUN4, MTERF4, GTPBP7, and the unfolded rRNA.

The online version of this article includes the following figure supplement(s) for figure 1:

**Figure supplement 1.** Cryo-EM analysis of human mitoribosomes Data processing scheme.

**Figure supplement 2.** Gold-standard FSC curve (**A**) and local resolution (**B**) of the mtLSU assembly intermediate map. FSC, Fourier shell correlation; mtLSU, mitoribosomal large subunit.

introduced mutations at locations in the three factors that we predicted would disrupt binding to the mtLSU without affecting expression, mitochondrial targeting or protein folding (*Figure 3A*, *Figure 3—figure supplement 1*, *Figure 3—figure supplement 2*).

Mutations that disrupt mitoribosome binding to MTER-4 (MTER-4$^{R178E,K262E,R263E}$) and MTG-1 (MTG-1$^{R171E,K178E,K180E}$; the *C. elegans* orthologs of MTERF4 and GTPBP7, respectively), led to a substantial increase in the expression of the mitochondrial heat-shock protein HSP-6, consistent with activation of the mitochondrial unfolded protein response (*Figure 3A*). These findings are similar to the effects observed when mitochondrial ribosomal proteins are knocked down by RNAi (*Houtkooper et al., 2013*). By contrast, we found that mutations that disrupt the catalytic activity of NSUN-4 did not substantially activate *hsp-6* expression (*Figure 3A*, NSUN-4 cd), nor did mutations that disrupt a weak contact between NSUN-4 and the mitoribosomal protein bL33m, NSUN-4$^{M225A, K226A}$. These results are consistent with the catalytic activity of NSUN-4 being dispensable for mitochondrial protein synthesis. However, NSUN-4 makes multiple contacts with the mitoribosome and the chosen M225A and K226A substitutions in NSUN-4 were likely insufficient to disrupt its interaction with the mitoribosome. We therefore introduced the additional mutations L286Y and A287Y, resulting in a dramatic mitochondrial stress response (*Figure 3A*, compare *nsun-4*$^{M225A,K226A}$ and *nsun-4*$^{M225A,K226A,L286Y,A287Y}$). Therefore, disrupting NSUN-4, MTER-4, or MTG-1 binding to the conserved core of the mitoribosome potently activates mitochondrial stress, underscoring their importance during mitoribosome biogenesis.

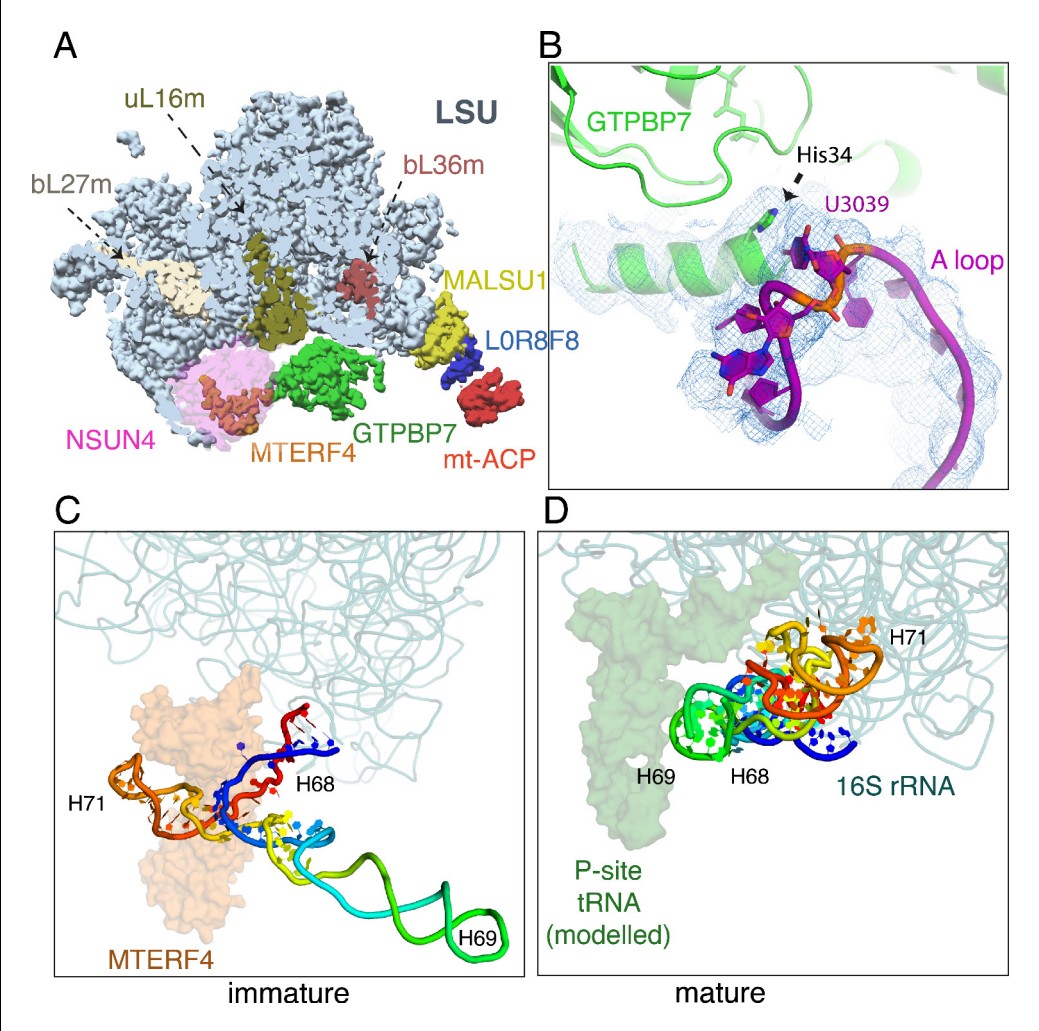

**Figure 2.** Maturation of the mtLSU by NSUN4·MTERF4·GTPBP7. (**A**) Incorporation of uL16m (olive), bL27m (wheat) precedes NSUN4 (magenta) binding and before bL36m (brown). NSUN4 is shown translucent for clarity. (**B**) His34 of helix 1 of GTPBP7 verifies that 2′-O-methylation of A-loop base U3039 by MRM2 has occurred. Immature (**C**) and mature (**D**) conformations of domain IV rRNA helices H68–71 (rainbow). H69 would pack against a canonical P-site tRNA when present, and together this region forms the back wall of the A, P, and E sites of the mature 39S. MTERF4 (translucent orange) holds H68–71 in an unfolded conformation to permit assembly factors to access the core of the mtLSU. mtLSU, mitoribosomal large subunit; rRNA, ribosomal RNA.

The online version of this article includes the following figure supplement(s) for figure 2:

**Figure supplement 1.** Comparison of MTG1-bound human mtLSU and RbgA-bound *Bacillus subtilis* 45S.

**Figure supplement 2.** GTPBP7 binds directly to the A loop of the 16S rRNA.

---

To test whether these mutations weaken 39S binding and to rule out the possibility that the observed mitochondrial stress phenotypes result from off-pathway consequences such as gene silencing, protein misfolding, or mistargeting, we sought to quantify the total- and ribosome-bound NSUN-4, MTER-4, and MTG-1 levels in mutant strains relative to wild-type worms. Given the absence of antibodies against these proteins, we had to resort to a suitable alternative method of quantification. Tandem mass tag-mass spectrometry (TMT-MS) permits accurate quantification of changes in global protein levels between multiple strains in parallel (*Zhang and Elias, 2017*). Total protein extracts as well as total ribosomes (cytosolic and mitochondrial) from wild-type, *nsun-4*[M225A, K226A] or *mter-4*[R178E,K262E,R263E] worms were extracted and labeled in duplicate (yielding a total of 12 samples) using isobaric tags and analyzed by TMT-MS (*Figure 3—figure supplement 3*, *Supplementary file 3*).

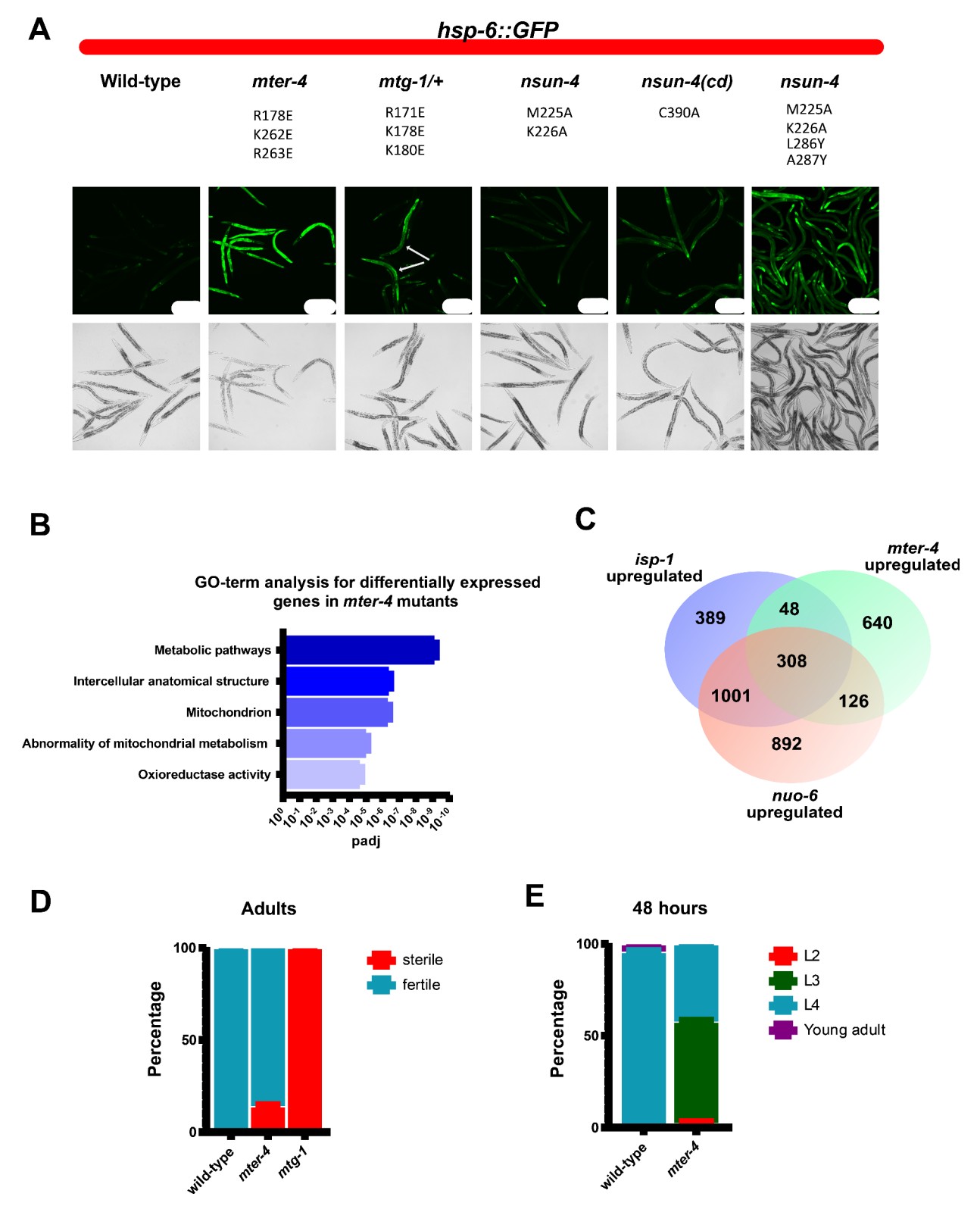

**Figure 3.** Mutations that disrupt mitoribosomal LSU binding result in activation of the mitochondrial unfolded protein response, delayed development, and decreased fertility. (**A**) Representative images of hsp-6::GFP expression in wild-type, *mter-4*[R178E,K262E,K263E], *mtg-1*[R171E,K178E,K180E], *nsun-4*[M225A,K226A], catalytic dead *nsun-4*[C90A], and *nsun-4*[M225A,K226A,L286Y,A287Y] mutant animals. The matched wild-type control for *nsun-4*[M225A,K226A,L286Y,A287Y] resembles the shown wild-type, and is omitted for simplicity. White arrows in mtg-1 mutants represent homozygous mutant animals based on the absence of

*Figure 3 continued on next page*

*Figure 3 continued*

umnIs21 reporter expression from balancer chromosome. All animals express the *zcIs13 [hsp-6::GFP]* reporter which is specifically expressed in response to mitochondrial stress. Animals were grown for 72 hr at 22.5°C. Scale bars=200 μm. (**B**) Top five GO-terms based on p-value from g:Profiler analysis of genes upregulated greater than 0.5 (log2) in *mter-4*$^{R178E,K262E,K263E}$ mutants when compared to wild-type animals. (**C**) Venn diagram comparison of genes upregulated in *mter-4*$^{R178E,K262E,K263E}$, *isp-1(qmv150)*, and *nuo-6(qm200)* mutants. *isp-1* and *nuo-6* gene expression data from **Yee et al., 2014**. (**D**) Fraction of wild-type and *mter-4*$^{R178E,K262E,K263E}$ mutants sterile at 22.5°C. n=100 animals. (**E**) Fraction of wild-type and *mter-4*$^{R178E,K262E,K263E}$ mutants at different developmental stages after 48 hr at 22.5°C. n=200 animals. GO, gene ontology.

The online version of this article includes the following figure supplement(s) for figure 3:

**Figure supplement 1.** Sequence alignments of *Caenorhabditis elegans* (Ce) and human (Hs) MTERF4 (top), NSUN4 (middle), and GTPBP7 (MTG1, bottom).

**Figure supplement 2.** Structural conservation of human and *Caenorhabditis elegans* MTERF4, NSUN4, and GTPBP7.

**Figure supplement 3.** TMT-MS quantification of protein levels (top).

The data sets were normalized between samples on cytosolic ribosomal protein levels following the reasonable assumption that manipulations to mitochondrial ribosome assembly/recycling factors do not affect cytosolic ribosomal protein levels. The results show that the levels of MTG-1, NSUN-4, and MTER-4, as well as the ribosomal proteins eL13 and mL53, are comparable to wild-type worms (1.05- to 1.57-fold; *Supplementary file 3*) in both *nsun-4*$^{M225A,K226A}$ and *mter-4*$^{R178E,K262E,R263E}$ worms (*Figure 3—figure supplement 3*, top), arguing against gene silencing, misfolding, or mistargeting of the designed mutants. This conclusion is also supported by the well-folded structures predicted by AlphaFold 2 (*Figure 3—figure supplement 2*; *Jumper et al., 2021*).

Mutating the mitoribosome binding surfaces of NSUN-4 or MTER-4 also significantly reduced their levels in the total ribosome fraction (63% and 86%, respectively) in *mter-4*$^{R178E,K262E,R263E}$ worms. As expected, the reductions were more modest (84% and 98%, respectively) in *nsun-4*$^{M225A,K226A}$ worms, which harbor more conservative mutations (*Figure 3—figure supplement 3*, bottom) and therefore also exhibit subtler stress responses (*Figure 3A*). More dramatic reductions would require all three proteins to be mutated simultaneously in the same strain, which was not done because these individual mutants already cause severe developmental defects, sterility, and poor viability. Taken together, the TMT-MS results confirm that the mitochondrial stress phenotypes are a direct consequence of disrupting the NSUN-4·MTER-4·MTG-1 complex and that these proteins play a crucial checkpoint role during mitoribosome biogenesis.

At the organismal level, destabilizing 39S binding by MTG-1 and MTER-4 resulted in delayed animal development and sterility (*Figure 3B,C*). Specifically, we found that *mter-4* mutants exhibited delayed development and 14% of mutants were sterile (*Figure 3B,C*). Similarly, we observed a more severe phenotype in *mtg-1* mutants and found that 100% of homozygous *mtg-1* mutants were sterile (*Figure 3B*). The severity of the phenotypes of the *nsun-4*, *mter-4*, and *mtg-1* mtLSU binding mutants underscores the importance of the role of NSUN4/NSUN-4, MTERF4/MTER-4, and GTPBP7/MTG-1 in the mitochondrion. In addition, our results suggest that the effects of disrupting GTPBP7/MTG-1 and MTERF4/MTER-4 interactions with the mitoribosome can be overcome in somatic tissues, potentially by activating the mitochondrial unfolded protein response, as mutant animals can indeed develop to adulthood. Germ cells, however, were particularly sensitive to these mutations, and disruption of mtLSU binding caused sterility. Notably, mutations in mitochondrial translational components, including mitochondrial tRNAs, are known to cause sterility in humans (*Demain et al., 2017*). We suggest that germ cell sensitivity to disruptions in mitochondrial translation is a conserved phenomenon throughout metazoans and establish a new model to study these effects.

To further confirm the effects of disrupting the interaction of these proteins with the mitoribosome, we performed next-generation RNA sequencing on the viable *mter-4* mutants. We identified 3174 differentially expressed genes (DEGs) in *mter-4* mutants when compared to wild-type animals (*Supplementary file 1*). Furthermore, we found that the genes upregulated in *mter-4* mutants are enriched for genes associated with mitochondrial dysfunction and substantially overlap with genes that are upregulated in electron transport chain mutants (*Yee et al., 2014*; *Figure 3D,E*). These results support our structural data and further suggest that disrupting the interaction of these proteins with the mitoribosome activates the mitochondrial unfolded protein response and disrupts normal mitochondrial function.

## Discussion

We have previously reported the cryo-EM structures of two human late-stage mitoribosomal assembly intermediates (*Brown et al., 2017*). One intermediate (PDB 5OOM) lacks bL36m, contains the anti-association factors MALSU1·L0R8F8·mt-ACP, and awaits folding of ~20% of the 16S rRNA, including the entire PTC. The second intermediate (PDB 5OOL) comprises fully matured 16S rRNA and all ribosomal proteins, but is incompetent for subunit joining because the anti-association factors are still present. Several events must therefore occur around these intermediates, and the order of events can be inferred from the reasonable assumption that assembly proceeds outwards from the core of the mtLSU to the inter-subunit interface (*Video 1*). (a) bL36m must bind the mtLSU and pack against the folded domain V helices H89–93; (b) the only remaining domain II helices H34–35 must fold; (c) H65–67 of domain IV can also then fold; and finally, (d) H68–71 of domain IV can fold to yield a mature PTC. PTC maturation also includes the universally conserved and essential modifications to incorporate pseudouridine Ψ3067 by RNA pseudouridine synthase (RPUSD4) and 2′-O-methylation at Gm2815, Um3039, and Gm3040 by mitochondrial methyltransferases1-3 (MRM1-3), respectively. These modifications likely happen in concert with the above steps.

A comprehensive review of GTPases in human mitoribosomal assembly (*Maiti et al., 2021*) and four studies describing cryo-EM structures of mtLSU late assembly intermediates distinct from ours and from each other (*Cheng et al., 2021*; *Cipullo et al., 2021*; *Hillen et al., 2021*; *Lenarčič et al., 2021*), appeared during manuscript preparation. All five studies, including ours, have trapped MALSU1·L0R8F8·mt-ACP and NSUN4·MTERF4 at the same location on the mtLSU. The studies, however, differ notably in the precise step during assembly, the biogenesis factors recruited, and finally, the GTPase in control. The deleterious consequences of manipulating NSUN4 or MTERF4 binding to the mitoribosome in vivo illustrate the importance of these proteins in mediating not only the exact step during PTC maturation that we have trapped, but also the prior and subsequent steps described in *Cheng et al., 2021*; *Cipullo et al., 2021*; *Hillen et al., 2021*.

When taken together, these cryo-EM structures permit piecing together the order of events during the late-stage maturation of the mtLSU (*Figure 4*). The anti-association factors MALSU1, L0R8F8, and mt-ACP engage the mtLSU during early assembly of the mtLSU (*Brown et al., 2017*; *Cheng et al., 2021*). At this stage, ribosomal proteins bL27m and uL16m have already been incorporated (state *I*). GTPBP10, MRM3, and DDX28 hold the PTC helices H89–93 and the central protuberance H80–88 in an immature state (*Figure 4*, state *II*) (*Cheng et al., 2021*). Following dissociation of these factors, the central protuberance matures and bL33m and bL35m are incorporated to result in state *III* (*Brown et al., 2017*; *Cheng et al., 2021*).

MTERF4 binds to and holds open H68–71 in an unfolded state and recruits NSUN4 to the mtLSU (*Cámara et al., 2011*). The ribosomal protein bL36m is incorporated, and MRM2 and GTPBP5 also then bind to 2′-O-methylate U3039 in the A loop and assist with H89–93 maturation (state *IV*) (*Cipullo et al., 2021*). Curiously, GTPBP7 is also present in state *IV*, but at a different location than in our structure. The close proximity of helices H34–35 and H65–67 to this location leads us to propose that GTPBP7 binds here to ensure that these helices have folded. GTPBP6, whose binding to the mtLSU is incompatible with the presence of NSUN4 (*Hillen et al., 2021*), displaces NSUN4 and GTPBP5 and helps to further mature the PTC (state *V*).

Next, GTPBP7, which was not actually seen in state *V* (*Hillen et al., 2021*), dissociates and repositions to the location on the PTC seen in our study (state *VI*). It then re-recruits NSUN4•MTERF4 and together NSUN4·MTERF4·GTPBP7 unfold H68–71 and check U3039 methylation and proper maturation of the PTC by the actions in the prior states *III–V*. This hypothesis if correct, would be consistent with (i) the dual roles of GTPBP6 during mitoribosome biogenesis and recycling (*Hillen et al., 2021*;

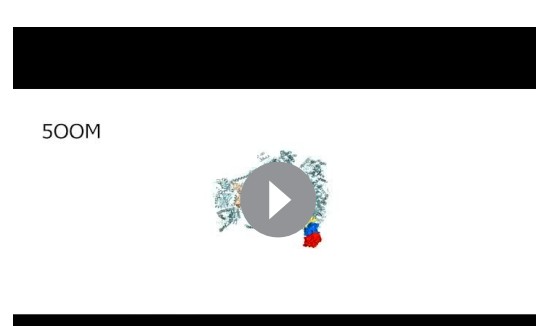

5OOM

**Video 1.** Proposed events during formation of the mitoribosomal peptidyl transferase centre.
https://elifesciences.org/articles/68806#video1

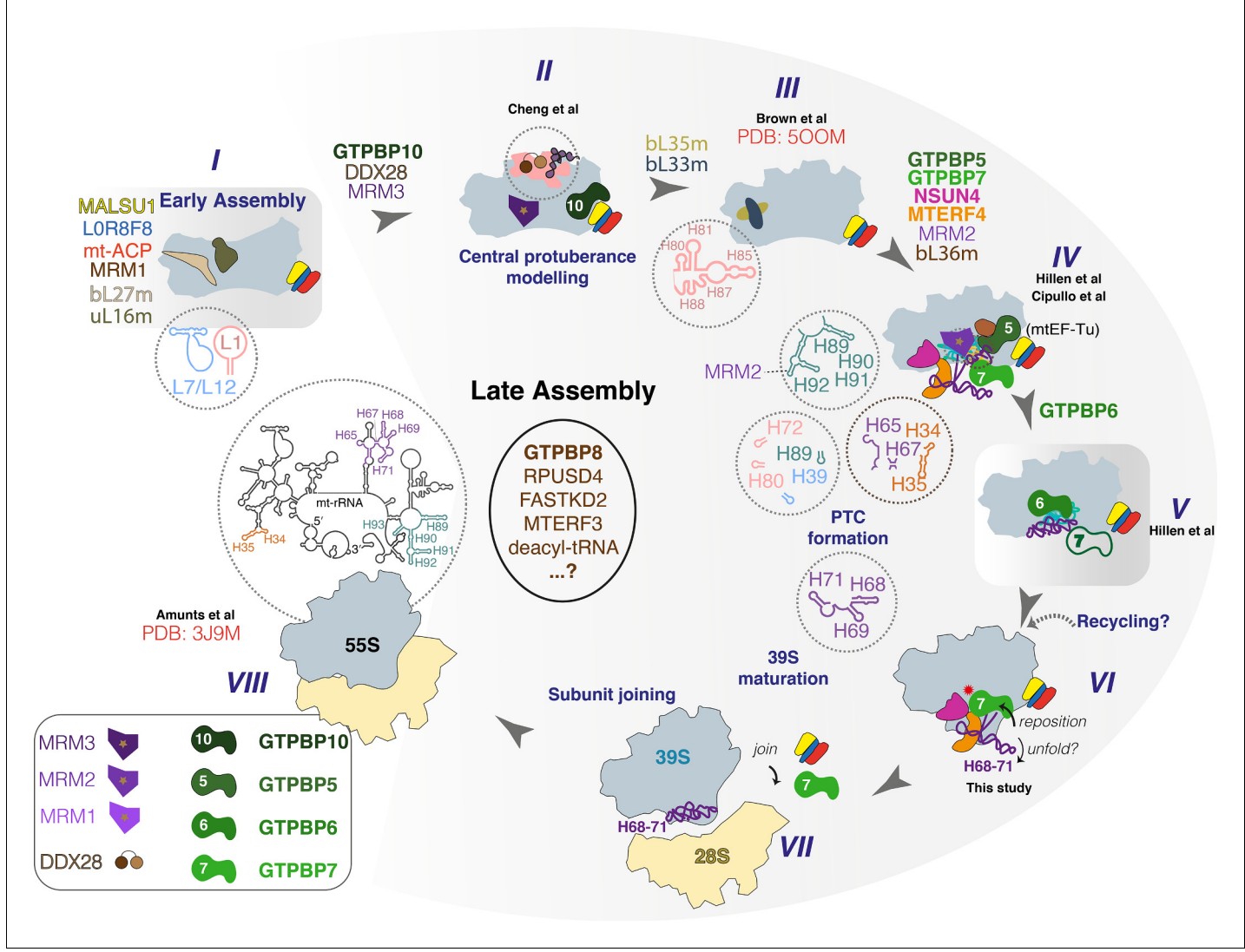

**Figure 4.** Proposed sequence of events in late-stage mtLSU assembly. States I–VIII along the human mtLSU late-stage maturation pathway, including recently reported states II (*Cheng et al., 2021*), III (*Brown et al., 2017*), IV (*Cipullo et al., 2021*; *Hillen et al., 2021*), V (*Hillen et al., 2021*), VI (this study), and VIII (*Amunts et al., 2015*). Note that state V was seen only to contain GTPBP6 and GTPBP7 is therefore depicted hollow. mtLSU, mitoribosomal large subunit.

*Lavdovskaia et al., 2020*), and (ii) the intriguing idea that not only GTPBP6 (*Hillen et al., 2021*) and MALSU1·L0R8F8·mt-ACP (*Desai et al., 2020*), but also NSUN4·MTERF4·GTPBP7 may have dual roles during mitoribosome recycling after termination and perhaps rescue.

This idea is supported by the facts that (i) we trapped our intermediate from Pde12 knockout (KO) cells which are a model for ribosome stalling (and subsequent recycling) (*Desai et al., 2020*; *Pearce et al., 2017*), (ii) during mitoribosome recycling, GTPBP6 rearranges the PTC of the 39S to a slightly immature state that could require subsequent checking by GTPBP7 (*Hillen et al., 2021*; *Lavdovskaia et al., 2020*), (iii) GTPBP7 is also involved in 39S–28S subunit joining (*Kim and Barrientos, 2018*; *Maiti et al., 2021*).

Given that we trapped GTPBP7 using the non-hydrolyzable GMPPCP, GTPBP7 may then dissociate upon GTP hydrolysis, eject NSUN4·MTERF4, thus allowing H68–71 to refold into the PTC and mediate subunit joining to result in a mature 55S (state *VIII*) (*Amunts et al., 2015*). How the anti-association factors dissociate prior to subunit joining is not yet known (*Kim and Barrientos, 2018*; *Maiti et al., 2021*).

The pathological significance of GTPBP7, MTERF4, and NSUN4 is demonstrated by their association with cardiomyopathy, which is a common feature of mitochondrial diseases. Cardiomyocyte-specific MTERF4 KO mouse models develop mitochondrial cardiomyopathy, and global KO mice are embryonically lethal (*Cámara et al., 2011*). Variant alleles of MTERF4 have also been documented in a pediatric patient with hypertrophic cardiomyopathy (*Maribel et al., 2018*). Similarly, GTPBP7 and NSUN4 silencing in human cardiomyocytes (*Kim and Barrientos, 2018*) and conditional KO mice have been shown to lead to impaired cardiac physiology and progressive cardiomyopathy (*Metodiev et al., 2014*), respectively.

Our observations of severe mitochondrial defects and activation of the mitochondrial unfolded protein response in *mter-4* mutants, but not *nsun-4* catalytic dead mutants, suggest that the major function of these proteins is as a checkpoint in mitochondrial ribosome assembly and not in RNA methylation. Consistent with this idea, MTERF4 does not participate during NSUN4-mediated methylation of the decoding center of the mtSSU (*Cámara et al., 2011*). We have thus shown, using structure-directed mutagenesis in vivo, that while the methyltransferase activity of NSUN4 on the small subunit can be dispensed with without resulting in a strong phenotype, the protein has a more important non-enzymatic function as a biogenesis factor in the final stages of the formation of the PTC and the tRNA binding sites in the large subunit.

The proper folding of the PTC, the universally conserved active site of the ribosome, along with the tRNA binding sites, is one of the crucial steps in the assembly of the large subunit. We conclude that this assembly occurs in multiple steps, moving from one in which the elements of the PTC are in a highly disordered state to one where they are ordered but not yet finally folded into their final structure in the large subunit. This assembly requires several protein factors to act together to yield the mature ribosomal subunit. By coupling cryo-EM analysis of human mitoribosome assembly intermediates with in vitro and in vivo studies in *C. elegans*, we have identified the physiological importance of three factors in an essential, conserved late-stage mtLSU maturation checkpoint. Although our studies were specifically on mitoribosomes, given the universally conserved nature of the PTC and its importance to the ribosome, we expect that other domains of life will have a similarly complex and orchestrated process requiring multiple factors to facilitate the formation of the active sites of the large ribosomal subunit.

## Materials and methods

### Cell line

PDE12$^{-/-}$ HEK293T cells that were used have previously been published (*Pearce et al., 2017*). Initially, cell growth was adherently at 37°C, 5% $CO_2$, in 10% fetal bovine serum (FBS) supplemented with Dulbecco's modified Eagle's medium (DMEM). Subsequently, the cells were adapted to grow in suspension in 1% FBS supplemented Freestyle media at 37°C, 8% $CO_2$.

### Purification of native mitoribosomal complexes

Human mitochondria, and from them native mitoribosomal complexes, were purified from PDE12$^{-/-}$ HEK293T cells as previously described (*Desai et al., 2020*). Briefly, purified cells were harvested at $10^6$ cells/ml and washed in cold phosphate-buffered saline (PBS). The weighed pellet of cells was resuspended in 6 ml of MIB buffer (50 mM HEPES-KOH pH 7.5, 10 mM KCl, 1.5 mM $MgCl_2$, 1 mM EDTA, 1 mM EGTA, 1 mM DTT, Proteinase Inhibitor; PI, 1 tablet per 50 ml) per gram and SM4 buffer (281 mM sucrose, 844 mM mannitol in MIB buffer) to yield 70 mM sucrose and 210 mM mannitol. The cells were subjected to nitrogen cavitation for 20 min at 500 psi to disrupt the cell membranes and the collected sample was centrifuged two times at 800×g for 15 min and the supernatant was collected. Two additional centrifugation steps at 10,000×g for 15 min were performed and each time, the pellet was resuspended in 0.5 ml of MIBSM buffer (3:1 MIB:SM4) per 1 g of originally weighed pellet. Per 1 g of the original pellet 10U of RNase-free DNase was added and rotated on a roller at 4°C for 20 min. The solution was then pelleted at 10,000×g for 15 min, resuspended in SEM buffer (250 mM sucrose, 20 mM HEPES·KOH pH 7.45, and 1 mM EDTA), dounce-homogenized, and layered onto a sucrose step gradient of 60%, 32%, 23%, and 15% sucrose for centrifugation at 99,004×g for 1 hr at 4°C. The collected mitochondrial layer was flash-frozen in liquid nitrogen for storage at −80°C.

To isolate mitoribosomes from the collected mitochondrial fraction, 2 volumes of lysis buffer (25 mM HEPES·KOH pH 7.4, 100 mM KCl, 25 mM Mg(OAc)$_2$, 1.5% β-DDM, 0.15 mg/ml TOCL, 0.5 mM GMPPCP (Sigma-Aldrich), 1 tablet per 50 ml PI, and 2 mM DTT) were added and briefly dounce-homogenized and stirred for 30 min in the cold room. Supernatant was collected after centrifugation at 30,000×g for 30 min and loaded at a ratio of 2.5:1 onto a 1 M sucrose cushion (20 mM HEPES·KOH, pH 7.4, 100 mM KCl, 20 mM Mg(OAc)$_2$, 0.6% β-DDM, 0.06 mg/ml TOCL, 0.25 mM GMPPCP, and 2 mM DTT) and centrifuged at 231,550×g for 60 min at 4°C. The pellet was resuspended (20 mM HEPES·KOH, pH 7.4, 100 mM KCl, 5 mM Mg(OAc)$_2$, 0.3% β-DDM, 0.03 mg/ml TOCL, 0.25 mM GMPPCP, and 2 mM DTT) and the crude mitoribosome fraction was layered onto a 15–30% sucrose gradient and subjected to centrifugation at 4°C for 90 min at 213,626×g. Following sucrose gradient fractionation, the mitoribosomal fractions were pooled and concentrated into final buffer (20 mM HEPES·KOH, pH 7.4, 100 mM KCl, 5 mM Mg(OAc)$_2$, 0.05% β-DDM, 0.005 mg/ml TOCL, 0.25 mM GMPPCP, and 2 mM DTT) before the mitoribosomes were vitrified for cryo-EM.

## Grid preparation and data collection

Quantifoil R2/2 holey carbon grids, covered with homemade amorphous carbon (~50 Å thick) was used with prior glow discharging. Using the Vitrobot Mk IV, 3 µl of our sample was applied to the grids, blotted for 4–6 s, vitrified by plunging into liquid ethane, and stored in liquid nitrogen. Data was collected over seven separate sessions in integrating mode at a magnification of 75,000, pixel size 1.04 Å on the FEI Titan Krios 300 kV electron microscope using the FEI Falcon III detector and EPU software. A total of 46,109 movies were collected using a defocus range from −1.1 to −3.2 at a dose of 1.5 e$^-$ per frame per Å$^2$ at 1 s exposure (39 frames).

## Image processing

RELION-3.0 and 3.1 (*Zivanov et al., 2018*) were used to process cryo-EM data as described before (*Desai et al., 2020*). Resolutions are reported according to the Fourier shell correlation=0.143 criterion (*Rosenthal and Henderson, 2003*). Movie motion correction was performed using MotionCorr2 (*Zheng et al., 2017*) and the contrast transfer function was estimated using CTFFIND-4.1 (*Rohou and Grigorieff, 2015*). 3,374,367 particles were picked from 43,950 micrographs that contained data to better than 6 Å resolution using a 2D reference (*Figure 1—figure supplement 1*). Micrographs with a CTF figure of merit >0.3 and a maximum resolution better than 5 Å were selected for further processing. Following particle extraction (128-pixel box; 5.0375 Å/pixel) and 2D classification, 3,247,481 particles were retained. The first 3D refinement yielded a 10.2 Å map at Nyquist resolution, using a 60 Å lowpass-filtered reference mitoribosome (EMD-2876) (*Amunts et al., 2015*). The reported 3D refinements are based on gold-standard estimates. A general 3D classification without alignments separated out mtLSU subclasses consisting of 1,283,454 particles. Following refinement and further 3D classification without alignment, a class of 1,100,599 particles was obtained that had previously unmodeled cryo-EM density at the subunit interface. This class was re-extracted to a pixel size of 1.04 Å and refined to 3.1 Å. Two subsequent rounds of focussed classification with signal subtraction were performed using generous soft masks (with 11-pixel extensions and 7-pixel cosine soft edges) that spanned the P and E sites and L1 stalk and A, P, and E sites, respectively. Following 3D refinement, CTF refinement and Bayesian polishing, the LSU assembly intermediate class was refined to 3.4 Å and post-processed in RELION and also using the highRes training model in deepEMhancer (*Sanchez-Garcia et al., 2020*).

## Model building, refinement, and validation

PDB 7A5F (*Desai et al., 2020*) was used as a starting model for model building in Coot v.0.9.3 (*Casañal et al., 2020*). Real-space refinement and validation were performed using phenix.real_space_refine (*Afonine et al., 2018*) and the Phenix suite (*Adams et al., 2010*), respectively. Homology models for MTERF4, NSUN4, and GTPBP7 were generated using trRosetta (*Yang et al., 2020*) and corrected using AlphaFold 2 (*Jumper et al., 2021*). The unfolded rRNA helices 68–71 were modeled using RNAcomposer (*Biesiada et al., 2016*) using secondary structure predictions from RNAfold (*Lorenz et al., 2011*). Only the backbone coordinates but not the bases were retained for the unstructured regions of H68–71. Comparisons were made with PDB 5OOL and 5OOM (*Brown et al., 2017*).

## *Caenorhabditis elegans* alleles and strain maintenance

All strains were grown and maintained at 20°C on NGM agar plates seeded with *Escherichia coli* HB101 unless otherwise stated. Individual point mutations in *mter-4(syb3662, syb3403)*, *mtg-1 (syb3641)*, and *nsun-4(syb3514)* were generated by SunyBiotech (Fuzhou, China). *nsun-4(syb3514)* results in an M225A and K226A substitution in NSUN-4, whereas nsun-4(syb3514, syb4263) additionally carries the mutations L286Y and A287Y. *mtg-1(syb3641)* results in R171E, K178E, and K180E substitutions in MTG-1. *mter-4(syb3403)* results in an R178E substitution in MTER-4. *mter-4(syb3662)* results in an K262E and R263E substitution in MTER-4. *nsun-4(mj457)* results in a C390A catalytic dead conversion in NSUN-4 and was reported previously (*Navarro et al., 2021*). All mutants were crossed with SJ4100—*zcls13[hsp-6::GFP]*—to generate *hsp-6* reporter strains. CGC32 - [*sC1(s2023) [dpy-1(s2170) umnIs21] III*] - was used to balance *mtg-1(syb3641)* which resulted in sterility.

## *Caenorhabditis elegans* imaging

Animals were collected as embryos and grown on NGM agar plates seeded with *E. coli* HB101 at 22.5°C. Adult animals were collected and immobilized in tetramisole and imaged using a Leica DM6 B and a Leica DFC9000 GT camera.

## RNA-seq

Animals were collected as embryos and grown on NGM agar plates seeded with *E. coli* HB101 at 20°C. Young adults were collected and washed three times in M9 buffer and snap-frozen in liquid nitrogen. Pellets of animal tissue were frozen and thawed five times and then refrozen at −70°C. RNA extraction and sequencing were performed by BGI Genomics (Hong Kong, China). Briefly, RNA was extracted by phenol-chloroform extraction. mRNA molecules were purified from total RNA using oligo(dT)-attached magnetic beads. cDNA was generated using random hexamer-primed reverse transcription. Libraries for paired-end 100 bp DNBSeq were generated and validated on the Agilent Technologies 2100 bioanalyzer.

## RNA-seq data analysis

Raw reads were trimmed for adaptors, low-quality sequences, and short reads with Trimmomatic (version 0.39, parameters: ILLUMINACLIP:TruSeq3-SE.fa:2:30:10 SLIDINGWINDOW:4:20 MIN-LEN:20) (*Bolger et al., 2014*). From the trimmed reads, the remaining rRNA was removed with sortmeRNA (version 2.1, default parameters) (*Kopylova et al., 2012*). Clean reads were then mapped to the *C. elegans* reference genome (WBCEL235) with HISAT2 (*Kim et al., 2019*) (version 2.1.0, default parameters) and raw counts for each were produced with HTSeq-count (*Anders et al., 2015*). Other quality control metrics were obtained with fastQC, Picard Tools, and multiQC (*Ewels et al., 2016*). Counts were imported into R and differential gene expression analysis was performed with DESeq2 (FDR<0.01, LFC>| 0.5 |) (*Love et al., 2014*). Gene ontology analysis of DEGs was performed with gProfiler (*Raudvere et al., 2019*).

## Developmental rate and sterility assays

Animals were collected as embryos and grown on NGM agar plates seeded with *E. coli* HB101 at 22.5°C. To assay developmental rate 200 animals were scored for their developmental stage at 48 hr. To assay for sterility 100 L4 stage animals were each individually transferred to a new plate. Animals with progeny after 4 days were scored as fertile. Animals with no progeny after 4 days were scored as sterile.

## Tandem mass tag-mass spectrometry

### Sample preparation

500 µl of packed wild-type, *mter-4(syb3662)* and *nsun-4(syb3514)* adult worms that were grown at 20°C were collected and stored at −80°C. Frozen pellets were thawed in 250 µl of either RNC buffer (50 mM HEPES pH 7.4, 100 mM KOAc, 5 mM Mg(OAc)$_2$, and 1× RNASIN RNAse inhibitor) or denaturing buffer (100 mM HEPES pH 7.6, 8 M urea) for total ribosome and total protein harvesting, respectively. 0.8 g of 0.7 mm diameter zirconia beads were added and the samples lysed in a QIAGEN TissueLyser (30 Hz, 2.5 min) at 4°C. The lysate was clarified by centrifugation at 16,100×g for 30 min at 4°C and the supernatant was either quantified and directly used for TMT-MS (in denaturing

buffer) or total ribosomes were isolated by centrifugation at 100,000 rpm at 4°C for 1 hr in a TLA 100.3 rotor through a 20% sucrose cushion in RNC buffer and the ribosome pellet resuspended in denaturing buffer. All samples were analyzed in biological duplicates.

### Enzymatic digestion

Protein samples in denaturing buffer were reduced with DTT and alkylated with iodoacetamide. Samples were then diluted to 4 M urea and digested with Lys-C (Promega). After 4 hr, the samples were further diluted to 1.6 M urea and digested overnight with trypsin (Promega) at 30°C. Digestion was stopped by the addition of formic acid (FA) to a final concentration of 0.5% and the resulting samples were desalted using homemade C18 stage tips (3M Empore) filled with poros R3 (Applied Biosystems) resin. Bound peptides were eluted with 30–80% MeCN/0.5% FA and lyophilized.

### Tandem mass tag labeling

Dried peptide mixtures from each condition were re-suspended in 200 mM Hepes, pH 8.5. TMT 10-plex reagent (Thermo Fisher Scientific), reconstituted according to manufacturer's instructions, and incubated at 23°C for an hour. The labeling reaction was then terminated by incubation with 5% hydroxylamine for ½ hr. The labeled peptides were pooled into a single sample and acetonitrile was removed by Speed Vac (Savant). Each set of TMT samples was desalted and pH 8 fractionated using the same stage tips method as above. Bound peptides were eluted with a 7.5–60% MeCN gradient in 0.1% triethylamine and fractionated into eight fractions. Eluted fractions were acidified, partially dried down in Speed Vac, and ready for LC-MS/MS.

### Mass spectrometry analysis

The fractionated peptides were analyzed by LC-MS/MS using a fully automated Ultimate 3000 RSLC nano System (Thermo Fisher Scientific) fitted with a 100 µm×2 cm PepMap100 C18 nano trap column and a 75 µm×25 cm, nanoEase M/Z HSS C18 T3 column (Waters). Peptides were separated using a binary gradient consisting of buffer A (2% MeCN, 0.1% formic acid) and buffer B (80% MeCN, 0.1% formic acid). Eluted peptides were introduced directly via a nanospray ion source into an Orbitrap Eclipse Tribrid mass spectrometer (Thermo Fisher Scientific). The mass spectrometer was operated data-dependent mode, performed MS1 scan (m/z=400–1600, resolution 120k) followed by MS2 acquisitions with a resolution of 50k, NCE of 38, and an isolation window set at 0.4 Th. Dynamic exclusion was set for 60 s.

### Proteome discoverer

The acquired MSMS raw files were processed using Proteome Discoverer (version 2.4, Thermo Fisher Scientific). MSMS spectra were searched against *C. elegans* proteome, UniProt Fasta database (downloaded in 2020), using Sequest search engine. Carbamidomethylation of cysteines, TMT610plex (N-term) and TMT10plex (K) were set as fixed modifications, while methionine oxidation and N-terminal acetylation (protein) were selected as variable modifications. The abundance values of TMT reporter ions were normalized to a set of mitoribosomal or alternatively, cytosolic ribosomal proteins. The output file from Proteome Discoverer, the proteins table was filtered for proteins with FDR of 1% and exported as excel files.

## Figure preparation

Adobe Illustrator 2021, UCSF Chimera (*Pettersen et al., 2004*), Pymol (Schrödinger, LLC), and Geneious 10.2.6 (https://www.geneious.com) were used for figure preparations.

## Statistics and reproducibility

Sample sizes for experiments involving *C. elegans* were selected based on similar studies from the literature and all animals from each genotype were selected and analyzed randomly. All replicate numbers listed in figure legends represent biological replicates of independent animals cultured separately, collected separately, and analyzed separately. Fisher's exact test was used to calculate p-values for *Figure 3b and c*.

Source data for all graphs can be found in the Statistics Source Data Table (*Supplementary file 2*).

## Acknowledgements

The authors thank Jake Grimmett and Toby Darling for advice, data storage, and high-performance computing; Michal Minczuk for providing the experimental cell line; Sew-Yeu Peak-Chew for TMT-mass spectrometric analysis; the Ramakrishnan lab members for useful discussions and reagents and Archana Yerra for discussions and writing support and Alan Brown for critical reading of the manuscript. The authors acknowledge the MRC Laboratory of Molecular Biology Electron Microscopy Facility for access and support of electron microscopy, sample preparation, and data collection. This work was supported by the UK Medical Research Council MC_U105184332, a Wellcome Trust Senior Investigator award (WT096570), the Agouron Institute, and the Louis-Jeantet Foundation to VR, and Cancer Research UK (C13474/A18583, C6946/A14492) and the Wellcome Trust (219475/Z/19/Z, 092096/Z/10/Z) to EAM ND is funded by a Wellcome Trust Clinical PhD Fellowship (110301/Z/15/Z). NB is funded by a Next Generation Research Fellowship at the Centre for Trophoblast Research. HY is funded by an EMBO Long-term Fellowship (EMBO ALTF 806–2018).

## Additional information

### Funding

| Funder | Grant reference number | Author |
| --- | --- | --- |
| Medical Research Council | MC_U105184332 | V Ramakrishnan |
| Wellcome Trust | WT096570 | V Ramakrishnan |
| Agouron Institute | | V Ramakrishnan |
| Louis-Jeantet Foundation | | V Ramakrishnan |
| Cancer Research UK | C13474/A18583 | Eric A Miska |
| Wellcome Trust | 219475/Z/19/Z | Eric A Miska |
| Cancer Research UK | C6946/A14492 | Eric A Miska |
| Wellcome Trust | 092096/Z/10/Z | Eric A Miska |

The funders had no role in study design, data collection and interpretation, or the decision to submit the work for publication.

### Author contributions

Viswanathan Chandrasekaran, Conceptualization, Formal analysis, Validation, Investigation, Visualization, Methodology, Writing - original draft, Project administration, Writing - review and editing; Nirupa Desai, Formal analysis, Funding acquisition, Validation, Investigation, Visualization, Methodology, Writing - original draft, Writing - review and editing; Nicholas O Burton, Conceptualization, Resources, Data curation, Formal analysis, Funding acquisition, Validation, Investigation, Visualization, Methodology, Writing - original draft, Writing - review and editing; Hanting Yang, Data curation, Funding acquisition, Investigation, Methodology, Writing - review and editing; Jon Price, Data curation, Formal analysis, Investigation, Methodology; Eric A Miska, Supervision, Funding acquisition, Project administration, Writing - review and editing; V Ramakrishnan, Resources, Supervision, Funding acquisition, Project administration, Writing - review and editing

### Author ORCIDs

Viswanathan Chandrasekaran (iD) https://orcid.org/0000-0002-0871-4740
Nirupa Desai (iD) http://orcid.org/0000-0001-6046-653X
Nicholas O Burton (iD) http://orcid.org/0000-0002-5495-3988
Hanting Yang (iD) http://orcid.org/0000-0002-3383-2204
Jon Price (iD) http://orcid.org/0000-0001-6554-5667
Eric A Miska (iD) http://orcid.org/0000-0002-4450-576X
V Ramakrishnan (iD) https://orcid.org/0000-0002-4699-2194

Decision letter and Author response
Decision letter https://doi.org/10.7554/eLife.68806.sa1
Author response https://doi.org/10.7554/eLife.68806.sa2

# Additional files

## Supplementary files

- Supplementary file 1. Next-generation RNA sequencing data.
- Supplementary file 2. Statistics source data.
- Supplementary file 3. TMT-MS source data.
- Transparent reporting form

## Data availability

Cryo-EM movies have been deposited to the EMPIAR database (EMPIAR-10809). Coordinates and maps have been deposited to the PDB (7PD3) and EMDB (EMD-13329), respectively. RNA-seq data is available through NCBI GEO using the accession code GSE169089 and as a supplementary item. Further information and material requests may be made to the corresponding authors.

The following datasets were generated:

| Author(s) | Year | Dataset title | Dataset URL | Database and Identifier |
|---|---|---|---|---|
| Chandrasekaran V, Desai N, Burton NO, Yang H, Price J, Miska EA, Ramakrishnan V | 2021 | Structure of the human mitoribosomal large subunit in complex with NSUN4. MTERF4.GTPBP7 and MALSU1.L0R8F8.mt-ACP | https://www.rcsb.org/structure/7PD3 | RCSB Protein Data Bank, 7PD3 |
| Price J, Burton NO, Miska EA | 2021 | mter-4 C. elegans RNA-seq | https://www.ncbi.nlm.nih.gov/geo/query/acc.cgi?acc=GSE169089 | NCBI Gene Expression Omnibus, GSE169089 |
| Desai N, Yang H, Chandrasekaran V, Kazi R, Minczuk M, Ramakrishnan V | 2021 | Single particle cryo-EM of human mitochondrial ribosomes from Pde12 KO cells | https://www.ebi.ac.uk/empiar/EMPIAR-10809/ | Electron Microscopy Public Image Archive, EMPIAR-10809 |
| Chandrasekaran V, Desai N, Burton NO, Yang H, Price J, Miska EA, Ramakrishnan V | 2021 | Structure of the human mitoribosomal large subunit in complex with NSUN4. MTERF4.GTPBP7 and MALSU1.L0R8F8.mt-ACP | https://www.ebi.ac.uk/pdbe/entry/emdb/EMD-13329 | Electron Microscopy Data Bank, EMDB-13329 |

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
