## [Decision Letter]

**Acceptance summary:**

Ribosomes are among the most complex molecular machines a cell makes. The work by Chandrasekaran et al., contributes to our understanding of the molecular details of mitochondrial ribosome assembly, and how disruptions to this pathway may cause human disease. Using cryo-EM, the authors identified a subpopulation of immature human mitochondrial large ribosomal subunits that interact with assembly factors NSUN4, MTERF4 and GTPBP7. Based on this structure, they introduce mutations in *C. elegans* orthologs of these assembly factors that are expected to disrupt binding to the large subunit, and they show that these mutants cause sterility and disrupt mitochondrial proteostasis in the mutant animals. They directly correlate the effect of mutating the mitoribosome binding surfaces on the *C. elegans* orthologs of NSUN-4 and MTER-4 with reduced co-migration with mitoribosomes. The magnitude of reduction in binding correlates well with the mutations and their ability to activate mitochondrial stress.

**Decision letter after peer review:**

Thank you for submitting your article "Visualising formation of the ribosomal active site in mitochondria" for consideration by *eLife*. Your article has been reviewed by 3 peer reviewers, including Nikolaus Grigorieff as Reviewing Editor and Reviewer #1, and the evaluation has been overseen by Cynthia Wolberger as the Senior Editor. The following individual involved in review of your submission has agreed to reveal their identity: Bronwyn A Lucas (Reviewer #2).

Essential revisions:

1) The authors found that mutations in the *C. elegans* orthologs of MTERF4 and GTPBP7 predicted from their model to interrupt the interaction with 16S rRNA cause sterility, mitochondrial proteotoxic stress and, for MTERF4, developmental delays. However, the authors did not confirm that these mutants interrupt the interaction with the LSU and that the mutant proteins were expressed at or above the level of their wild type counterparts. This makes it difficult to determine whether the observed physiological effects were due to loss of this interaction or down-regulation of the proteins. The authors should show that these mutations disrupt binding of MTERF4 and GTPBP7, and that the mutant proteins are produced at or above wild type levels.

2) The loss of mitochondrial proteostasis observed by the authors could have been caused by the mutant proteins themselves being misfolded, mistargeted or aggregated, as has been described for other ribosome binding proteins. These issues should be addressed in the text.

3) The authors should include a figure showing the H. sapiens interaction interfaces from their model indicating the resides that are or are not conserved in the *C. elegans* orthologs to support their argument of high conservation of these interfaces.

*Reviewer #1 (Recommendations for the authors):*

Discussion

1. The authors could strengthen their conclusions by discussion the possible artifacts introduced by the purification protocol and using GMPPCP to trap assembly intermediates. Since only about 2% of the collected data ended up in the final reconstruction, it may also be worth briefly discussing what the other 98% of the data represent, and how the authors decided that the selected 2% of the data represent the functional state they were seeking to investigate.

Grid preparation and Data collection

2. Please clarify that data collection was done using the Falcon detector in integrating mode.

Image Processing

3. Please cite Rosenthal and Henderson (2003), instead of Chen et al., (2013), for the FSC = 0.143 resolution threshold.

4. Please provide a reference for MotionCorr2.

5. Please describe how the 43,950 micrographs were selected from the 46,109 collected movies.

6. Please provide a reference for the lowpass-filtered reference mitoribosome used as an initial reference.

*Reviewer #2 (Recommendations for the authors):*

The authors would benefit from more a more cautious interpretation of their findings and a clearer delineation between description of results and speculation about implications.

Specific recommendations:

1. The authors should consider a more specific and descriptive title and avoid implying that the structure was determined in mitochondria.

2. It is well documented that RBP-RNA interactions can form post-lysis. The authors should include more discussion of the technical limitations of their approach in the interpretation of this structure.

3. Line 123: The authors should show that these mutations disrupt binding.

4. The authors should show that the mutant proteins are produced at or above wt levels.

5. In Figure 2 it is difficult to determine whether U3039 is 2' O methylated from the map as shown.

6. The authors do not show that 2' O methylation of U3039 is necessary for binding of GTPBP7.

7. U3039 and G3040 are listed as XX in the model. The authors note that G3040 was flexible and unable to be modeled. The authors should state why U3039 absent.

8. On lines 102-103 the authors' state that A loop "moves towards GTPBP7". The authors should state what this movement is relative to (an earlier structure? The mature mito-LSU?) and show this in Figure 2—figure supplement 2.

9. Line 114: define highly conserved, comment on the conservation of the interaction interface.

10. The two paragraphs on lines 162-179 should be moved to the discussion.

11. Are the labels for U3039 and G3040 in Figure 2—figure supplement 2 switched? The label "U3039" indicates a purine.

12. The authors should discuss how the interface that contacts the 16S rRNA compares in the related structures from Hillen et al., and Cipullo et al.,. Would the *C. elegans* Mtg-1 mutants be expected to disrupt this complex too?

13. Figure 3:B and C legends are reversed.

14. Developmental stage of Mtg-1 mutants is not displayed in Figure 3C. If not possible a justification should be provided.

15. The Mtg-1 mutant R178E is inconsistent with the protein sequence given. Are the author's referring to R179? Or K178?

16. The terms "GTPBP7" and "MTG1" are used interchangeably. E.g.: in Figure 2—figure supplement 1, both terms are used.

17. Figure 4: Add the word "proposed" to the figure legend title.

18. Navarro et al., 2020 is listed twice in the references.

*Reviewer #3 (Recommendations for the authors):*

In the recent past, several high-resolution cryo-EM structures of mitochondrial ribosomes (mitoribosomes) from different species have become available. The process by which these macromolecular complexes of eukaryotic ribosomes, in particular their mitoribosomes, assemble is poorly understood. However, high-resolution structures of maturation intermediates of these protein-rich mitoribosomes have just begun to emerge. The manuscript by Chandrasekaran and coworkers reports cryo-EM structures of human mitochondrial large (39S) ribosomal subunit in late assembly stages, in which the complexes were trapped using a non-hydrolysable analog of GTP. A variety of mitoribosomal complexes were isolated using extensive classification of the very large cryo-EM dataset. The major functional findings derived from the analyses of the 55S monosomal population from the same dataset was published last year by this group. As is usually the case, the mammalian mitoribosomal preparations carry a large pool of dissociated 39S subunit, a detailed analysis of 39S fraction is presented as part of this study. Authors identify six proteins that play specific roles in the 39S assembly process, three of which are positioned near the PTC of the 39S subunit, and apparently are involved in proper late-stage folding of the PTC's component rRNA, essential for the peptidyl transferase activity. In addition, authors correlate their structural findings with genetic and biochemical studies using *C. elegans* as a model organism. The study will be of interest to researcher working in the fields of translation, mitoribosome assembly, and in understanding the molecular mechanism of diseases associated with defects in mitoribosomal assembly. However, there is a major point that should be addressed, before the manuscript could be accepted for publication.

The comparison of structural work with human 39S subunit assembly intermediates with results of point mutations in C.elegans orthologs of the assembly factors provide a general biological link but lack direct structural correlation or comparison. A figure visualizing the locations (contact sites) of mutated *C. elegans* amino acids in the corresponding mammalian factors will help in a better understanding of the structural and functional correlations derived in the paper. For example, it could explain how the NSUN4 mutants described in line 133-134 could still be associated with the 39S subunit.

---

## [Author Response]

Essential revisions:1) The authors found that mutations in the *C. elegans* orthologs of MTERF4 and GTPBP7 predicted from their model to interrupt the interaction with 16S rRNA cause sterility, mitochondrial proteotoxic stress and, for MTERF4, developmental delays. However, the authors did not confirm that these mutants interrupt the interaction with the LSU and that the mutant proteins were expressed at or above the level of their wild type counterparts. This makes it difficult to determine whether the observed physiological effects were due to loss of this interaction or down-regulation of the proteins. The authors should show that these mutations disrupt binding of MTERF4 and GTPBP7, and that the mutant proteins are produced at or above wild type levels.

We agree that quantifying the levels of the mutant proteins is important, but this assay is very difficult to do in the absence of antibodies against the poorly characterized *C. elegans* orthologs. We have therefore addressed these two questions by performing tandem mass tag-mass spectrometry (TMT-MS) on urea-solubilized total protein extracts to quantify the overall expression levels of MTER-4 and NSUN-4 in the mutant worms relative to wild-type worms. The results (Figure 3 —figure supplement 3) reveal that the mutants are expressed at comparable levels to the wild-type proteins (1.0 – 1.6-fold), indicating that they are not turned over or silenced by cells, a fate that often occurs to misfolded or over-expressed protein variants.

In parallel, we also isolated total ribosomes (cytosolic + mitochondrial) from these worms in non-denaturing, physiological buffer and compared the amounts of MTER-4 and NSUN-4 bound to mitoribosomes. As expected, complexes carrying these mutations associated with mitoribosomes at only 63 – 86% of wild-type levels. Note that more dramatic reductions would require all three proteins to be mutated simultaneously in the same strain, which was not done because these individual mutants already cause severe developmental defects, sterility and poor viability.

Finally, the mutations were designed to disrupt solvent exposed sidechains at the mitoribosome binding interface and are not likely therefore to disrupt protein folding. This fact is again confirmed by AlphaFold 2-predicted structures of these mutants (Figure 3 —figure supplement 2).

2) The loss of mitochondrial proteostasis observed by the authors could have been caused by the mutant proteins themselves being misfolded, mistargeted or aggregated, as has been described for other ribosome binding proteins. These issues should be addressed in the text.

To address this concern, we performed a TMT-MS experiment which clearly demonstrated that this is not the case. We have updated the text to discuss this result.

3) The authors should include a figure showing the H. sapiens interaction interfaces from their model indicating the resides that are or are not conserved in the *C. elegans* orthologs to support their argument of high conservation of these interfaces.

Agreed. The sequence conservation is now highlighted in Figure 3 —figure supplement 1 and the structural conservation is also shown now in a new figure, Figure 3 —figure supplement 2.

Reviewer #1 (Recommendations for the authors):Discussion1. The authors could strengthen their conclusions by discussion the possible artifacts introduced by the purification protocol and using GMPPCP to trap assembly intermediates. Since only about 2% of the collected data ended up in the final reconstruction, it may also be worth briefly discussing what the other 98% of the data represent, and how the authors decided that the selected 2% of the data represent the functional state they were seeking to investigate.

The dataset has been exhaustively classified and the majority of findings from the other 98% of the data are now published (Desai et al., Science 370(6520) pg.1105-1110). GMPPCP is a well-characterized analog of GTP that has been used successfully to trap biologically relevant intermediates in many contexts, including other on-pathway elongation intermediates as in this publication above. It is perhaps possible that 39S assembly intermediates bound to the other GTPases relevant to 39S assembly, such as GTPBP6, are also present at very low abundance in this dataset.

Grid preparation and Data collection2. Please clarify that data collection was done using the Falcon detector in integrating mode.

This information is now in the Methods section and we have also mentioned this in the revised text.

Image Processing3. Please cite Rosenthal and Henderson (2003), instead of Chen et al., (2013), for the FSC = 0.143 resolution threshold.

Thank you. Done.

4. Please provide a reference for MotionCorr2.

Done.

5. Please describe how the 43,950 micrographs were selected from the 46,109 collected movies.

The sentence now reads “3,374,367 particles were picked from a subset of 43,950 micrographs that contained data to better than 6 Å resolution using a 2D reference”.

6. Please provide a reference for the lowpass-filtered reference mitoribosome used as an initial reference.

Done. EMD-2876.

Reviewer #2 (Recommendations for the authors):The authors would benefit from more a more cautious interpretation of their findings and a clearer delineation between description of results and speculation about implications.Specific recommendations:1. The authors should consider a more specific and descriptive title and avoid implying that the structure was determined in mitochondria.

The title is now “Visualising formation of the active site in the mitochondrial ribosome”.

2. It is well documented that RBP-RNA interactions can form post-lysis. The authors should include more discussion of the technical limitations of their approach in the interpretation of this structure.

Our in vivo results, including the newly-performed TMT-MS argue strongly against this interpretation. We have updated the text to describe these new results, and their in vivo significance, in more detail.

3. Line 123: The authors should show that these mutations disrupt binding.

We have performed TMT-MS to verify that the mutating residues that mediate mitoribosome binding do reduce their levels in the ribosome fractions. Comparing the newly-added nsun-4^M225A,K226A,L286Y,A287Y^ mutant against the previous nsun-4^M225A,K226A^ mutant reveals that these mutations specifically weaken the conserved binding interface to the 39S and activate mitochondrial stress responses.

4. The authors should show that the mutant proteins are produced at or above wt levels.

The TMT-MS reveals that all three proteins are expressed at between 1 – 1.6-fold higher levels than the wild-type.

5. In Figure 2 it is difficult to determine whether U3039 is 2' O methylated from the map as shown.

We agree. We have replaced the original Figure 2B with an improved version.

6. The authors do not show that 2' O methylation of U3039 is necessary for binding of GTPBP7.

This was never a claim. We suspect that although this is possible, it is more likely that GTP hydrolysis and egress of GTPBP7 is triggered only when the helix containing His34 binds the conformation that Um3039 adopts upon methylation. This explanation would be more consistent with a checkpoint role for GTPBP7 and would allow GTPBP7 to sample the A loop repeatedly until Mrm2 methylates U3039.

7. U3039 and G3040 are listed as XX in the model. The authors note that G3040 was flexible and unable to be modeled. The authors should state why U3039 absent.

This is likely an issue with the specific software used. The PDB deposition server appears to recognize modified bases correctly. We have also now modelled both residues.

8. On lines 102-103 the authors' state that A loop "moves towards GTPBP7". The authors should state what this movement is relative to (an earlier structure? The mature mito-LSU?) and show this in Figure 2—figure supplement 2.

The figure has been updated and the sentence now reads: “Our structure reveals that the A and P loops moves towards GTPBP7 and NSUN4 respectively relative to their positions in the mature 55S (Figure 2—figure supplement 2) and His 34 of helix 1 of GTPBP7 directly contacts the highly conserved U3039 to verify 2’-O-methylation by mitochondrial methyltransferase 2 (MRM2)”.

9. Line 114: define highly conserved, comment on the conservation of the interaction interface.

The % identities of the homologs are now mentioned parenthetically. The interaction interface comprises the universally conserved core of the mitoribosome, and this is now mentioned.

10. The two paragraphs on lines 162-179 should be moved to the discussion.

This is a good idea.

11. Are the labels for U3039 and G3040 in Figure 2—figure supplement 2 switched? The label "U3039" indicates a purine.

Yes, thank you for spotting the error.

12. The authors should discuss how the interface that contacts the 16S rRNA compares in the related structures from Hillen et al., and Cipullo et al., Would the *C. elegans* Mtg-1 mutants be expected to disrupt this complex too?

We have added the following to the Discussion. Our in vivo analysis is also applicable to Hillen et al., Cipullo et al. and Cheng et al., (2021).

“The deleterious consequences of manipulating NSUN4 and MTERF4 in vivo illustrate the importance of these proteins in mediating not only the exact step during PTC maturation that we have trapped, but also the prior and subsequent steps described in (Cheng et al., 2021; Cipullo et al., 2021; Hillen et al., 2021).”

13. Figure 3:B and C legends are reversed.

Thank you. This is now fixed.

14. Developmental stage of Mtg-1 mutants is not displayed in Figure 3C. If not possible a justification should be provided.

The Mtg-1 mutants are not viable as homozygotes, and the analysis is complicated by the balancing that is done to compensate for this. We therefore simply chose only the mutant with the strongest phenotype (mter-4) to perform this quantification. All data from RNA-seq experiments was performed on young adult animals and this information is now in the Methods section.

15. The Mtg-1 mutant R178E is inconsistent with the protein sequence given. Are the author's referring to R179? Or K178?

K178E. Thank you.

16. The terms "GTPBP7" and "MTG1" are used interchangeably. E.g.: in Figure 2—figure supplement 1, both terms are used.

This is now fixed. We chose to retain the *C. elegans* nomenclature of MTG-1 and the human name of GTPBP7.

17. Figure 4: Add the word "proposed" to the figure legend title.

We have added the word “proposed”.

18. Navarro et al., 2020 is listed twice in the references.

This is now fixed.

Reviewer #3 (Recommendations for the authors):In the recent past, several high-resolution cryo-EM structures of mitochondrial ribosomes (mitoribosomes) from different species have become available. The process by which these macromolecular complexes of eukaryotic ribosomes, in particular their mitoribosomes, assemble is poorly understood. However, high-resolution structures of maturation intermediates of these protein-rich mitoribosomes have just begun to emerge. The manuscript by Chandrasekaran and coworkers reports cryo-EM structures of human mitochondrial large (39S) ribosomal subunit in late assembly stages, in which the complexes were trapped using a non-hydrolysable analog of GTP. A variety of mitoribosomal complexes were isolated using extensive classification of the very large cryo-EM dataset. The major functional findings derived from the analyses of the 55S monosomal population from the same dataset was published last year by this group. As is usually the case, the mammalian mitoribosomal preparations carry a large pool of dissociated 39S subunit, a detailed analysis of 39S fraction is presented as part of this study. Authors identify six proteins that play specific roles in the 39S assembly process, three of which are positioned near the peptidyl transferase center (PTC) of the 39S subunit, and apparently are involved in proper late-stage folding of the PTC's component rRNA, essential for the peptidyl transferase activity. In addition, authors correlate their structural findings with genetic and biochemical studies using *C. elegans* as a model organism. The study will be of interest to researcher working in the fields of translation, mitoribosome assembly, and in understanding the molecular mechanism of diseases associated with defects in mitoribosomal assembly. However, there is a major point that should be addressed, before the manuscript could be accepted for publication.The comparison of structural work with human 39S subunit assembly intermediates with results of point mutations in C.elegans orthologs of the assembly factors provide a general biological link but lack direct structural correlation or comparison. A figure visualizing the locations (contact sites) of mutated *C. elegans* amino acids in the corresponding mammalian factors will help in a better understanding of the structural and functional correlations derived in the paper. For example, it could explain how the NSUN4 mutants described in line 133-134 could still be associated with the 39S subunit.

We have now added a figure (Figure 3—figure supplement 2) to illustrate the mutations. The newly added NSUN4^M225A,K226A,L286Y,A287Y^ mutant weakens mitoribosome binding better than does NSUN4^M225A,K226A^ and also activates mitochondrial stress more potently.